# CrossSparse-MoE: Adaptive Sparsity and Cross-Channel Expert Routing for Time Series Forecasting

## Abstract

Time series forecasting under limited data remains challenging due to model overfitting and insufficient structural regularization. In this work, we uncover a sparsity-oriented scaling phenomenon: as training data increases, model parameters naturally become sparser—even in simple linear models. This observation motivates the introduction of learned sparsity as an effective prior to improve model generalization under data-scarce regimes. We propose CrossSparse-MoE, a lightweight forecasting framework that enhances model expressiveness while promoting adaptive sparsity. Built upon a linear backbone, CrossSparse-MoE incorporates cross-channel convolutions to capture short-term inter-variable dependencies and employs a Mixture-of-Experts (MoE) module with non-linear MLPs. A learnable gating network dynamically routes temporal segments to specialized experts, while L1 regularization encourages parameter sparsity without imposing rigid structural constraints. Extensive experiments on multiple benchmarks demonstrate that CrossSparse-MoE consistently outperforms state-of-the-art baselines, particularly in low-data scenarios, validating the effectiveness of combining structural flexibility with learned sparsity. Code is available in Appendix.

## 1 Introduction

Time series forecasting is fundamental to a wide range of real-world applications, including energy management, financial modeling, and industrial monitoring. Recent progress has been largely driven by deep learning methods, including Transformer-based models (Zhou et al., 2021; Wu et al., 2022; Zhou et al., 2022; Nie et al., 2023; Lin et al., 2023a), CNN-based models (Liu et al., 2022; Wu et al., 2023; Wang et al., 2022), and MLP-based architectures (Xu et al., 2024; Ekambaram et al., 2023; Das et al., 2023; Huang et al., 2024a). These models have achieved impressive performance, particularly in long-term forecasting (LTSF) tasks. However, most of them rely heavily on large-scale, high-quality labeled datasets. In practice, such data is often limited, leading to overfitting and poor generalization—especially in cross-domain or few-shot scenarios (Chang et al., 2024; Jin et al., 2024; Wang et al., 2025).

To improve model efficiency and robustness under limited data, many recent works have explored sparse and lightweight architectures (Lin et al., 2024; Shi et al., 2025; Chen et al., 2024; Ni et al., 2024; Ismail et al., 2023; Zeng et al., 2023; Zhang et al., 2022). These approaches enforce sparsity through explicit architectural design—such as block-wise pruning (Lin et al., 2024), temporal gating (Shi et al., 2025), and frequency-domain filtering (Xu et al., 2024)—to reduce computation and overfitting. While effective, they impose fixed structural constraints that may limit model flexibility and prevent sparsity from adapting to dataset-specific patterns.

In contrast, we revisit an underexplored but fundamental property in time series models: *parameter sparsity*. Through empirical investigation, we uncover a novel sparsity-oriented scaling law—model parameters become naturally sparser as training data increases, even without explicit regularization. This observation suggests that sparsity can emerge as a learned inductive bias and provides a promising direction for improving generalization in low-resource forecasting.

Motivated by this, we propose **CrossSparse-MoE**, a hybrid and lightweight forecasting framework that balances model capacity and adaptive sparsity. It consists of two key components: (1)

a cross-channel convolutional embedding to enhance short-term inter-variable modeling, and (2) a Mixture-of-Experts module composed of multiple non-linear experts with dynamic gating. To encourage sparse and interpretable representations, we apply $\ell_1$ regularization to expert weights, enabling pruning of redundant parameters.

With our meticulously designed architecture, our CrossSparse-MoE achieves state-of-the-art performance on various long-term time series forecasting tasks, while maintaining a lightweight design that offers superior efficiency and speed compared to more complex TSF methods under limited computational resources.

Our contributions are summarized as follows:

- We uncover a *sparsity-oriented scaling law* in time series forecasting: model parameters naturally become sparser as training data increases.
- We propose **CrossSparse-MoE**, a novel forecasting framework that combines a cross-channel MoE architecture with $\ell_1$ regularization.
- We demonstrate state-of-the-art performance and efficiency across diverse benchmarks, highlighting the generalization ability of adaptive sparsity.

## 2 RELATED WORKS

**Long Time Series Forecasting.** Time series forecasting has seen significant progress with deep learning models, which can be broadly categorized into univariate and multivariate approaches. Univariate models such as DeepState (Rangapuram et al., 2018), DeepAR (Salinas et al., 2020), and N-BEATS (Oreshkin et al., 2020) focus on individual time series, while multivariate models are designed to handle multiple correlated sequences simultaneously. Transformer-based architectures have played a central role in recent advancements, especially in long-term forecasting (LTSF) tasks. Early works like Informer (Zhou et al., 2021), Autoformer (Wu et al., 2022), and FEDformer (Zhou et al., 2022) modified the standard Transformer to better capture temporal patterns. More recent models such as PatchTST (Nie et al., 2023) and PETformer (Lin et al., 2023a) show that the vanilla Transformer, when equipped with patching strategies inspired by computer vision (Dosovitskiy et al., 2020; He et al., 2022), can achieve strong performance. Beyond Transformers, CNN- and MLP-based models—e.g., SCINet (Liu et al., 2022), TimesNet (Wu et al., 2023), MICN (Wang et al., 2022), TiDE (Das et al., 2023), and HDMixer (Huang et al., 2024a)—have demonstrated that simpler architectures can be competitive. Additionally, RNN-based models such as SegRNN (Lin et al., 2023b) and graph-based models like CrossGNN (Huang et al., 2024b) have been revisited for LTSF, showing promising results. Recently, the adaptation of pretrained large language models (LLMs) to time series forecasting (Chang et al., 2024; Jin et al., 2024; Xue & Salim, 2023) has opened new directions, though challenges remain in generalization, particularly under cross-domain, few-shot, or zero-shot settings (Wang et al., 2025).

**Sparse and Lightweight Modeling in Time Series Forecasting.** Recent studies have increasingly focused on improving the efficiency of time series forecasting models through sparsity and lightweight design. SparseTSF (Lin et al., 2024) applies block-wise masking to prune temporal and channel dimensions, while Time-MoE (Shi et al., 2025) introduces temporal gating to selectively activate expert modules. Similarly, Pathformer (Chen et al., 2024), MoLE (Ni et al., 2024), and IME (Ismail et al., 2023) leverage expert routing or attention masking to enforce structured sparsity. Models like DLinear (Zeng et al., 2023), LightTS (Zhang et al., 2022), TSMixer (Ekambaram et al., 2023), and FITS (Xu et al., 2024) reduce parameter budgets through temporal mixing or frequency-domain filtering. Although these methods significantly improve computational and parameter efficiency, *the sparsity is manually imposed through architectural design rather than dynamically learned from data*. This limits their adaptability to diverse datasets and prevents parameter sparsity from emerging naturally during training.

In contrast, our proposed CrossSparse-MoE combines structural modularity with data-driven sparsity learning. It incorporates a cross-channel Mixture-of-Experts architecture and enforces parameter sparsity via L1 regularization. Moreover, we identify a *sparsity-oriented scaling law*, showing that parameters become inherently sparser as training data grows—a phenomenon not explored in prior time series models.

# 3 PARAMETER SPARSITY UNDER DATA SCALING: THEORY AND REGULARIZATION

 

Figure 1: Visualization of weight sparsification with increasing data volume at intervals of 24 on the ETTh2 dataset.

Figure 2: Distinctive pattern of weight changes corresponding to data volume variations on the weather dataset.

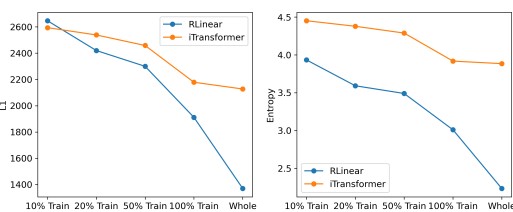

Figure 3: Visualization of Linear projection layer weights varying with data volume.

## 3.1 OBSERVATION

Figure 3.3 visualizes the weight matrices of a Linear model trained with varying amounts of data on ETTh2 and weather datasets. In Fig. 1, the ETTh2 dataset exhibits a clear trend: as training data increases from 10% to 100%, the weight matrices become increasingly sparse and structured, with prominent diagonal patterns emerging. This indicates that the model gradually focuses on essential channel interactions while suppressing redundant ones as more data is available. In contrast, Fig. 2 shows a more nuanced sparsity evolution on the weather dataset. Although sparsification still increases with data volume, the patterns are more complex and less diagonally dominant, reflecting fundamentally different temporal and inter-channel dependencies. These results highlight that sparsity and structural patterns in model weights emerge naturally with more training data, shaped by the underlying data characteristics.

To verify the generality of this phenomenon, we provide additional visualizations in the supplementary material across other datasets, almost showing similar trends. Furthermore, when training on the full dataset (training + validation + test), we observe even stronger sparsification (as shown in Fig. 3), suggesting that access to more diverse temporal patterns further enhances weight pruning.

## 3.2 MOTIVATION

While the primary training objective for time series forecasting models is typically the minimization of prediction loss (e.g., MSE), we observe a consistent empirical trend: *as the size of the training dataset increases, the learned model parameters become increasingly sparse*. Interestingly, this phenomenon arises without any explicit sparsity constraints in the optimization objective. In this section, we present a theoretical interpretation of this behavior and further propose a sparsity-aware regularization strategy to enhance model robustness under limited data.

## 3.3 DATA-INDUCED PARAMETER SPARSITY

We consider a forecasting model $f_\theta$ trained with the following objective:

$$\min_\theta \ \mathbb{E}_{(x,y)\sim\mathcal{D}} \left[ \|f_\theta(x) - y\|_2^2 \right], \tag{1}$$

where $\theta \in \mathbb{R}^p$ are the learnable parameters. While this formulation includes no explicit sparsity term, we find that when trained on sufficiently large datasets, the model naturally suppresses redundant parameters. This phenomenon can be attributed to the following mechanisms:

**Gradient Stabilization with Increasing Data.** Let $g_B(\theta) = \frac{1}{B}\sum_{i=1}^{B}\nabla_\theta \ell(f_\theta(x_i), y_i)$ denote the batch gradient at each step. Although the short-term noise in gradient estimation is governed by the batch size $B$, the *long-term gradient structure*—i.e., whether a parameter is consistently updated—depends on the full dataset size $|D|$. As $|D| \to \infty$, we have:

$$\mathbb{E}_{(x,y)\sim D}\left[\frac{\partial \mathcal{L}}{\partial \theta_j}\right] \to \mathbb{E}_{(x,y)\sim \mathcal{D}}\left[\frac{\partial \mathcal{L}}{\partial \theta_j}\right]. \tag{2}$$

Parameters with near-zero expected gradients across the dataset receive vanishing updates during training and are naturally suppressed. Thus, the model exhibits a form of *data-driven pruning*, where only truly predictive weights remain active.

**Implicit Bias Toward Low-Complexity Solutions.** In overparameterized settings, stochastic gradient descent is known to converge toward *minimum-norm solutions*. With ample data, the optimization landscape becomes flatter and more constrained, leading to convergence in regions of lower weight magnitude and structural redundancy. Even without explicit regularization, the optimizer implicitly favors sparse configurations to minimize complexity while retaining predictive capacity.

**Sparsity Scaling Law.** Empirically, we observe that the number of nonzero parameters $\|\theta\|_0$ decreases sublinearly with training data size $D$. We propose the following scaling relation:

$$\mathbb{E}_D\left[\|\theta\|_0\right] \le C_1 + C_2 \cdot D^{-\beta}, \quad \beta > 0, \tag{3}$$

indicating that sparsity emerges as an inductive consequence of data abundance. This behavior aligns with our broader understanding of neural scaling phenomena in time series forecasting (Shi et al., 2024).



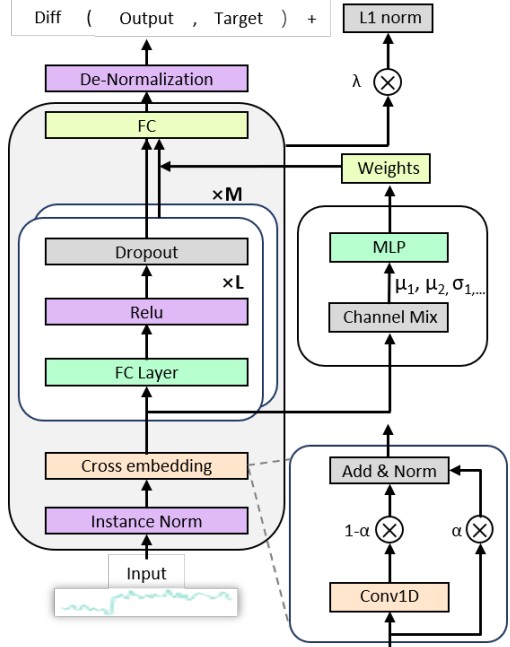

Figure 4: Visualization of Linear projection weights under different filter ratios (FR) and L1 regularization settings on the ETTh2 dataset. Top: full data, Bottom: 50% data; Left: without L1 regularization, Right: with L1 regularization. L1 norm encourages weight sparsity and enhances structural diagonality, especially under low-resource settings.

Figure 5: Overview of CS-MoE, including RevIN normalization, correlation-aware embedding, expert routing via gating, weighted expert fusion, and L1-based sparsity regularization.

### 3.4 SPARSITY-AWARE REGULARIZATION FOR LOW-RESOURCE SETTINGS

While data-induced sparsity naturally arises under large datasets, such behavior cannot be reliably expected in *low-resource regimes*, where the model lacks sufficient gradient consensus to suppress uninformative parameters. In such cases, we propose to explicitly inject sparsity bias into the learning objective via $\ell_1$-regularization.

**Modified Objective.** We augment the forecasting loss with a sparsity-inducing regularizer:

$$\mathcal{L}_{\text{total}}(\theta) = \mathcal{L}_{\text{forecast}}(\theta) + \lambda \cdot \|\theta\|_1, \tag{4}$$

where $\lambda > 0$ is a hyperparameter controlling the strength of the sparsity prior. This modification is equivalent to a MAP estimation under a Laplace prior and serves to guide the optimizer toward low-complexity solutions, especially when training data is insufficient to do so organically.

**Implementation Details.** In practice, we apply $\ell_1$-regularization to all learnable weights. The coefficient $\lambda$ is selected from $\{10^{-5}, 10^{-4}, 10^{-3}\}$ based on validation performance.

**Visual Evidence of L1-Induced Sparsity.** To intuitively demonstrate the effect of L1 regularization, we visualize the learned Linear projection weights under different filter ratios (FR) in Figure 4. The top row corresponds to full training data (FR=1.0), while the bottom row shows the case with only 50% data (FR=0.5). We compare results with and without L1 regularization.

We observe that without L1, the weight matrices are relatively dense and exhibit noisy patterns, especially under low-resource conditions. With L1, the matrices become notably sparser, and clear diagonal structures emerge, indicating more focused and structured channel interactions. This suggests that L1 serves as an effective regularization tool to promote both sparsity and interpretability, especially when the data scale is limited.

## 4 METHOD

We propose **CS-MoE**, a compact and adaptive forecasting model that integrates inter-channel convolution and a sparse mixture-of-experts framework to improve generalization under data-limited settings. The overall architecture is illustrated in Figure 5. CS-MoE consists of three main components: (1) correlation-aware convolutional embedding, (2) temporal-aware mixture-of-experts, and (3) L1-based expert regularization.

### 4.1 INSTANCE NORMALIZATION AND DE-NORMALIZATION

To handle temporal distribution shift and stabilize input dynamics, we adopt RevIN (Kim et al., 2021), a parameter-free instance normalization module. Given input $X \in \mathbb{R}^{B \times C \times T}$, we compute channel-wise statistics over the temporal axis:

$$\mu = \frac{1}{T} \sum_{t=1}^{T} X_{:,:,t}, \quad \sigma = \sqrt{\frac{1}{T} \sum_{t=1}^{T} (X_{:,:,t} - \mu)^2}. \tag{5}$$

The normalized input is:

$$X^{\text{norm}} = \frac{X - \mu}{\sigma}, \tag{6}$$

ensuring zero mean and unit variance per channel. After forecasting, we recover the original scale via:

$$\hat{Y} = X^{\text{denorm}} = \hat{Y}^{\text{norm}} \cdot \sigma + \mu. \tag{7}$$

### 4.2 CORRELATION-AWARE CHANNEL EMBEDDING

Most previous forecasting models either ignore inter-variable dependencies or impose strong structural priors (e.g., attention or graphs), which may increase overfitting risks under low-resource settings. To address this, we introduce a lightweight correlation-aware embedding module that captures

localized temporal correlations within each variable while preserving a simple and interpretable structure.

Formally, given an input sequence $X \in \mathbb{R}^{B \times C \times T}$, where $B$ is the batch size, $C$ is the number of variables (channels), and $T$ is the temporal length, we define the embedding as:

$$X^{\text{emb}} = \alpha \cdot X + (1 - \alpha) \cdot \text{Conv1D}_{\text{time}}(X), \tag{8}$$

where $\alpha \in \mathbb{R}$ is a learnable scalar initialized to 1.0, and $\text{Conv1D}_{\text{time}}$ denotes a 1D convolution with kernel size 3 applied independently to each channel along the temporal axis. Specifically, the convolution does not mix across channels, and no bias or activation function is applied.

The learnable parameter $\alpha$ allows the model to interpolate between the original input and the convolved signal. Since $\alpha$ is jointly optimized with the rest of the model, the network can adaptively decide the degree of residual blending based on task-specific patterns. In our implementation, we do not impose any explicit regularization or range constraint on $\alpha$, allowing the optimization to freely explore its effective value.

This design improves robustness by allowing localized temporal smoothing while avoiding overfitting to noise, especially under low-resource conditions.

### 4.3 TEMPORAL-AWARE MIXTURE-OF-EXPERTS

Our CS-MoE employs a mixture of $E$ temporal experts, each modeled as a nonlinear module that independently forecasts future sequences. To route inputs to appropriate experts, we compute a temporal summary vector via channel-wise averaging:

$$s = \text{Mean}_c(X) \in \mathbb{R}^{B \times T}. \tag{9}$$

This vector is fed into a lightweight gating network:

$$w = \text{Softmax}\left(\text{Linear}(\text{ReLU}(\text{Linear}(s)))\right) \in \mathbb{R}^{B \times E}, \tag{10}$$

producing expert selection weights. Each expert $E_i$ outputs a candidate forecast, and the final prediction is the weighted sum:

$$\hat{Y} = \sum_{i=1}^{E} w_i \cdot E_i(X), \tag{11}$$

where $E_i(X) \in \mathbb{R}^{B \times C \times L'}$ and $L'$ is the prediction horizon. This design promotes specialization and conditional computation. We apply $\ell_1$ regularization on expert parameters to enhance sparsity (see Sec. 3).

## 5 EXPERIMENTS

### 5.1 EXPERIMENT SETTINGS

**Datasets.** In line with previous studies (Qiu et al., 2024; Zhou et al., 2021; Jin et al., 2024; Nie et al., 2023), we evaluate our method on ten widely used real-world datasets that span a variety of application domains. These include datasets such as Electricity Transformer Temperature (ETT) (Zhou et al., 2021), Traffic, Electricity, Weather, National Illness (ILI) (Lai et al., 2018), and Exchange. All datasets are multivariate in nature, and we provide further details regarding their characteristics and preprocessing procedures both below and in the Appendix.

**Baselines.** To ensure fair and comprehensive evaluation, we compare our approach against a diverse set of recent state-of-the-art models. This includes convolution-based architectures such as Times-Net (Wu et al., 2023) and MICN (Wang et al., 2022); mixture-of-experts (MoE) models like MoLE (Mixture of Linear Experts) (Ni et al., 2024); MLP-based methods such as FITS (Xu et al., 2024), TimeMixer (Wang et al., 2024), and DLinear (Zeng et al., 2023); and Transformer-based methods including PDF (Dai et al., 2024) and PatchTST (Nie et al., 2023).

**Implementation Details.** All experiments are carried out on a workstation equipped with an NVIDIA GeForce RTX 3090 GPU running 64-bit Linux (kernel version 5.15.0-56-generic). For

the ETT and Solar datasets, we adopt a 60%/20%/20% split for training, validation, and testing, respectively, while a 70%/10%/20% split is applied to all other datasets. Following the protocol established in the TFB benchmark (Qiu et al., 2024), we perform a grid search over input sequence lengths in {96, 336, 512} to determine the optimal setting for each dataset.

| Models | Metric | CS-MoE MSE | CS-MoE MAE | MoLE MSE | MoLE MAE | PDF MSE | PDF MAE | FITS MSE | FITS MAE | TimeMixer MSE | TimeMixer MAE | PatchTST MSE | PatchTST MAE | MICN MSE | MICN MAE | DLinear MSE | DLinear MAE |
|---|---|---|---|---|---|---|---|---|---|---|---|---|---|---|---|---|---|
| ETTh1 | 96 | **0.357** | **0.387** | 0.375 | 0.390 | 0.360 | 0.391 | 0.376 | 0.396 | 0.372 | 0.401 | 0.377 | 0.397 | 0.378 | 0.412 | 0.379 | 0.403 |
| | 192 | 0.395 | **0.412** | 0.403 | 0.417 | **0.392** | 0.414 | 0.400 | 0.418 | 0.413 | 0.430 | 0.409 | 0.425 | 0.400 | 0.430 | 0.408 | 0.419 |
| | 336 | 0.424 | **0.427** | 0.430 | 0.434 | **0.418** | 0.435 | 0.419 | 0.435 | 0.438 | 0.450 | 0.431 | 0.444 | 0.428 | 0.447 | 0.440 | 0.440 |
| | 720 | **0.417** | **0.441** | 0.449 | 0.461 | 0.456 | 0.462 | 0.435 | 0.458 | 0.486 | 0.484 | 0.457 | 0.477 | 0.474 | 0.499 | 0.471 | 0.493 |
| | Avg | **0.398** | **0.417** | 0.414 | 0.425 | 0.407 | 0.426 | 0.408 | 0.427 | 0.427 | 0.441 | 0.419 | 0.436 | 0.420 | 0.447 | 0.425 | 0.439 |
| ETTh2 | 96 | **0.271** | **0.333** | 0.273 | 0.334 | 0.276 | 0.341 | 0.277 | 0.345 | 0.281 | 0.351 | 0.274 | 0.337 | 0.313 | 0.372 | 0.300 | 0.364 |
| | 192 | **0.330** | **0.373** | 0.336 | 0.374 | 0.339 | 0.382 | 0.331 | 0.379 | 0.349 | 0.387 | 0.348 | 0.384 | 0.419 | 0.439 | 0.387 | 0.423 |
| | 336 | 0.352 | **0.392** | 0.371 | 0.404 | 0.374 | 0.406 | **0.350** | 0.396 | 0.366 | 0.413 | 0.377 | 0.416 | 0.474 | 0.475 | 0.490 | 0.487 |
| | 720 | **0.380** | **0.420** | 0.409 | 0.439 | 0.398 | 0.433 | 0.382 | 0.425 | 0.401 | 0.436 | 0.406 | 0.441 | 0.723 | 0.600 | 0.704 | 0.597 |
| | Avg | **0.333** | **0.380** | 0.347 | 0.388 | 0.347 | 0.391 | 0.335 | 0.386 | 0.349 | 0.397 | 0.351 | 0.395 | 0.482 | 0.472 | 0.470 | 0.468 |
| ETTm1 | 96 | **0.284** | **0.331** | 0.291 | 0.333 | 0.286 | 0.340 | 0.303 | 0.345 | 0.293 | 0.345 | 0.289 | 0.343 | 0.303 | 0.349 | 0.300 | 0.345 |
| | 192 | **0.320** | **0.355** | 0.333 | 0.357 | 0.321 | 0.364 | 0.337 | 0.365 | 0.335 | 0.372 | 0.329 | 0.368 | 0.336 | 0.369 | 0.336 | 0.366 |
| | 336 | 0.357 | **0.381** | 0.368 | 0.383 | **0.354** | 0.383 | 0.368 | 0.386 | 0.368 | 0.386 | 0.362 | 0.390 | 0.370 | 0.391 | 0.367 | 0.386 |
| | 720 | **0.406** | **0.411** | 0.429 | 0.418 | 0.408 | 0.415 | 0.420 | 0.413 | 0.426 | 0.417 | 0.416 | 0.423 | 0.410 | 0.421 | 0.419 | 0.416 |
| | Avg | **0.342** | **0.369** | 0.355 | 0.373 | **0.342** | 0.376 | 0.357 | 0.377 | 0.356 | 0.380 | 0.349 | 0.381 | 0.355 | 0.383 | 0.356 | 0.378 |
| ETTm2 | 96 | **0.159** | **0.245** | 0.163 | 0.247 | 0.163 | 0.251 | 0.165 | 0.254 | 0.165 | 0.256 | 0.165 | 0.255 | 0.173 | 0.271 | 0.164 | 0.255 |
| | 192 | **0.213** | **0.281** | 0.217 | 0.286 | 0.219 | 0.290 | 0.219 | 0.291 | 0.225 | 0.298 | 0.221 | 0.293 | 0.232 | 0.313 | 0.224 | 0.304 |
| | 336 | **0.266** | **0.320** | 0.272 | 0.323 | 0.269 | 0.330 | 0.272 | 0.326 | 0.277 | 0.332 | 0.276 | 0.327 | 0.303 | 0.367 | 0.277 | 0.337 |
| | 720 | **0.349** | **0.376** | 0.380 | 0.391 | **0.349** | 0.382 | 0.359 | 0.381 | 0.360 | 0.387 | 0.362 | 0.381 | 0.467 | 0.477 | 0.371 | 0.401 |
| | Avg | **0.247** | **0.305** | 0.258 | 0.312 | 0.250 | 0.313 | 0.254 | 0.313 | 0.257 | 0.318 | 0.256 | 0.314 | 0.294 | 0.357 | 0.259 | 0.324 |
| Weather | 96 | **0.143** | **0.188** | 0.152 | 0.192 | 0.147 | 0.196 | 0.172 | 0.225 | 0.147 | 0.198 | 0.149 | 0.196 | 0.172 | 0.232 | 0.170 | 0.230 |
| | 192 | **0.186** | **0.224** | 0.190 | 0.228 | 0.193 | 0.240 | 0.215 | 0.261 | 0.192 | 0.243 | 0.191 | 0.239 | 0.214 | 0.270 | 0.216 | 0.275 |
| | 336 | **0.237** | **0.265** | 0.245 | 0.271 | 0.245 | 0.280 | 0.261 | 0.295 | 0.247 | 0.284 | 0.242 | 0.279 | 0.259 | 0.309 | 0.258 | 0.307 |
| | 720 | 0.310 | **0.318** | 0.316 | 0.324 | 0.323 | 0.334 | 0.326 | 0.341 | 0.318 | 0.330 | 0.312 | 0.330 | **0.309** | 0.343 | 0.323 | 0.362 |
| | Avg | **0.219** | **0.249** | 0.226 | 0.254 | 0.227 | 0.263 | 0.244 | 0.281 | 0.226 | 0.264 | 0.224 | 0.261 | 0.239 | 0.289 | 0.242 | 0.293 |
| Traffic | 96 | **0.362** | **0.238** | 0.372 | 0.246 | 0.368 | 0.252 | 0.400 | 0.280 | 0.369 | 0.257 | 0.370 | 0.262 | 0.517 | 0.313 | 0.395 | 0.275 |
| | 192 | **0.379** | **0.251** | 0.385 | 0.254 | 0.382 | 0.261 | 0.412 | 0.288 | 0.400 | 0.272 | 0.386 | 0.269 | 0.526 | 0.302 | 0.407 | 0.280 |
| | 336 | **0.391** | **0.258** | 0.407 | 0.266 | 0.393 | 0.268 | 0.433 | 0.308 | 0.407 | 0.272 | 0.396 | 0.275 | 0.545 | 0.307 | 0.417 | 0.286 |
| | 720 | 0.430 | 0.273 | **0.429** | **0.265** | 0.438 | 0.297 | 0.478 | 0.339 | 0.461 | 0.316 | 0.435 | 0.295 | 0.569 | 0.328 | 0.454 | 0.308 |
| | Avg | **0.391** | **0.255** | 0.398 | 0.258 | 0.395 | 0.270 | 0.431 | 0.304 | 0.409 | 0.279 | 0.397 | 0.275 | 0.539 | 0.313 | 0.418 | 0.287 |
| Electricity | 96 | **0.126** | **0.217** | 0.129 | 0.219 | 0.128 | 0.222 | 0.139 | 0.237 | 0.153 | 0.256 | 0.143 | 0.247 | 0.158 | 0.266 | 0.140 | 0.237 |
| | 192 | **0.143** | **0.235** | 0.150 | 0.237 | 0.147 | 0.242 | 0.154 | 0.250 | 0.168 | 0.269 | 0.158 | 0.260 | 0.175 | 0.287 | 0.154 | 0.251 |
| | 336 | **0.161** | **0.251** | 0.164 | 0.256 | 0.165 | 0.260 | 0.170 | 0.268 | 0.189 | 0.291 | 0.168 | 0.267 | 0.184 | 0.296 | 0.169 | 0.268 |
| | 720 | **0.187** | **0.276** | 0.188 | 0.278 | 0.199 | 0.289 | 0.212 | 0.304 | 0.228 | 0.320 | 0.214 | 0.307 | 0.200 | 0.310 | 0.204 | 0.301 |
| | Avg | **0.154** | **0.245** | 0.158 | 0.247 | 0.160 | 0.253 | 0.169 | 0.265 | 0.185 | 0.284 | 0.171 | 0.270 | 0.179 | 0.290 | 0.167 | 0.264 |
| Exchange | 96 | **0.079** | **0.198** | 0.084 | 0.201 | 0.083 | 0.200 | 0.082 | 0.199 | 0.084 | 0.207 | **0.079** | 0.200 | **0.079** | 0.203 | 0.080 | 0.202 |
| | 192 | 0.166 | 0.291 | 0.173 | 0.293 | 0.172 | 0.294 | 0.173 | 0.295 | 0.178 | 0.300 | 0.159 | **0.289** | **0.158** | 0.299 | 0.182 | 0.321 |
| | 336 | 0.311 | 0.402 | 0.321 | 0.407 | 0.323 | 0.411 | 0.317 | 0.406 | 0.376 | 0.451 | **0.297** | **0.399** | 0.300 | 0.403 | 0.327 | 0.434 |
| | 720 | 0.743 | 0.649 | 0.845 | 0.694 | 0.820 | 0.682 | 0.825 | 0.684 | 0.884 | 0.707 | 0.751 | 0.650 | 0.745 | 0.665 | **0.578** | **0.605** |
| | Avg | 0.325 | 0.386 | 0.356 | 0.399 | 0.350 | 0.397 | 0.349 | 0.396 | 0.381 | 0.416 | 0.322 | **0.385** | 0.321 | 0.393 | **0.292** | 0.391 |
| Solar | 96 | 0.172 | **0.196** | 0.186 | 0.203 | 0.181 | 0.247 | 0.208 | 0.255 | 0.179 | 0.232 | **0.170** | 0.234 | 0.190 | 0.250 | 0.199 | 0.265 |
| | 192 | **0.185** | **0.209** | 0.196 | 0.212 | 0.200 | 0.259 | 0.229 | 0.267 | 0.201 | 0.259 | 0.204 | 0.302 | 0.226 | 0.284 | 0.220 | 0.282 |
| | 336 | **0.187** | **0.213** | 0.194 | 0.215 | 0.208 | 0.269 | 0.241 | 0.273 | 0.190 | 0.256 | 0.212 | 0.293 | 0.259 | 0.308 | 0.234 | 0.295 |
| | 720 | **0.202** | 0.235 | 0.205 | **0.230** | 0.212 | 0.275 | 0.248 | 0.277 | 0.203 | 0.261 | 0.215 | 0.307 | 0.341 | 0.365 | 0.243 | 0.301 |
| | Avg | **0.186** | **0.213** | 0.195 | 0.215 | 0.200 | 0.263 | 0.232 | 0.268 | 0.193 | 0.252 | 0.200 | 0.284 | 0.254 | 0.302 | 0.224 | 0.286 |
| ILI | 24 | **1.635** | **0.782** | 1.752 | 0.821 | 1.801 | 0.874 | 2.182 | 1.002 | 1.804 | 0.820 | 1.932 | 0.872 | 2.279 | 1.020 | 2.208 | 1.031 |
| | 36 | **1.610** | **0.784** | 1.712 | 0.796 | 1.743 | 0.867 | 2.241 | 1.029 | 1.891 | 0.926 | 1.869 | 0.866 | 2.451 | 1.085 | 2.032 | 0.981 |
| | 48 | **1.512** | **0.777** | 1.731 | 0.813 | 1.843 | 0.926 | 2.272 | 1.036 | 1.752 | 0.866 | 1.891 | 0.883 | 2.440 | 1.077 | 2.209 | 1.063 |
| | 60 | **1.591** | **0.842** | 1.724 | 0.854 | 1.845 | 0.925 | 2.642 | 1.142 | 1.831 | 0.930 | 1.914 | 0.896 | 2.303 | 1.012 | 2.292 | 1.086 |
| | Avg | **1.587** | **0.796** | 1.730 | 0.821 | 1.808 | 0.898 | 2.334 | 1.052 | 1.820 | 0.886 | 1.902 | 0.879 | 2.368 | 1.049 | 2.185 | 1.040 |
| $1^{st}$ Count | | 81 | | 3 | | 5 | | 1 | | 0 | | 6 | | 3 | | 3 | |

Table 1: Fair long-term forecasting results under hyperparameter searching without the "drop-last" trick. The best model is **bold**, and the second best is underlined. Count is the number of the best results.

## 5.2 MAIN RESULTS

Table 1 presents a comprehensive comparison between our proposed method **CS-MoE** and a range of time series forecasting baselines across ten benchmark datasets under a fair long-term forecasting protocol. As clearly shown, CS-MoE achieves the **best performance in 81 out of all evaluated settings**, significantly outperforming existing state-of-the-art models such as MoLE, PDF, and PatchTST. Our model consistently ranks either first or second across nearly all datasets, highlighting its strong generalization ability. In particular, CS-MoE demonstrates superior forecasting accuracy in long-horizon settings, benefiting from its channel-sparse mixture-of-experts structure that enables both effective representation learning and efficient computation.

## 5.3 MODEL ANALYSIS

**Ablation study.** To better understand the contribution of each component in our CS-MoE, we conduct ablation study by removing or altering key modules. Table 2 summarizes the results averaged across all forecasting horizons on two representative datasets (ETTh2 and Weather). Removing the L1 norm regularization ("w/o L1 norm") leads to a notable degradation in performance, especially on the ETTh2 dataset, which confirms that our sparsity-inducing constraint plays a crucial role in improving generalization. Excluding the cross-channel embedding module ("w/o Cross Embed.") also results in performance drops, particularly on the Weather dataset, demonstrating the importance of capturing inter-channel dependencies. Lastly, replacing the Mixture-of-Experts structure with a single expert ("w/o MoE") leads to consistent decreases in accuracy across both datasets, highlighting the effectiveness of expert specialization within our framework. Overall, the full model (CS-MoE) achieves the best results across all metrics, validating the complementary benefits of its individual components.

| Variants | ETTh2 (avg.) | | Weather (avg.) | |
|---|---|---|---|---|
| | MSE | MAE | MSE | MAE |
| w/o L1 norm | 0.348 | 0.389 | 0.221 | 0.253 |
| w/o Cross Embed. | 0.341 | 0.387 | 0.233 | 0.265 |
| w/o MoE | 0.341 | 0.386 | 0.219 | 0.250 |
| **CS-MoE** | **0.333** | **0.379** | **0.218** | **0.249** |

Table 2: Ablation study. Results are averaged from all forecasting horizons $\in$ {96,192,336,720}.

| $\lambda$ | ETTh1 (avg.) | | Exchange (avg.) | |
|---|---|---|---|---|
| | MSE | MAE | MSE | MAE |
| 0.0 | 0.416 | 0.423 | 0.365 | 0.409 |
| 0.001 | 0.420 | 0.433 | 0.372 | 0.411 |
| 0.0001 | 0.404 | 0.419 | **0.338** | **0.392** |
| 0.00001 | **0.401** | **0.417** | 0.352 | 0.398 |
| 0.000001 | 0.412 | 0.421 | 0.360 | 0.405 |

Table 3: Effect of different L1 regularization coefficients $\lambda$ on forecasting performance. Results are averaged over all horizons on the ETTh1 and Exchange datasets.

**Efficiency Analysis** Table 5 compares the efficiency of different time series forecasting models in terms of computational cost (MACs), parameter size, inference time, and forecasting accuracy (MSE) on the large-scale Electricity dataset with a look-back window of 512 and a forecasting horizon of 720. Our model, CS-MoE, achieves superior efficiency, requiring only 560M MACs and 1.69M parameters—an order of magnitude lower than prior methods such as Pathformer and PDF. Additionally, CS-MoE delivers a significantly faster inference time of just 3.27ms, while also achieving the best prediction accuracy with an MSE of 0.189. These results highlight CS-MoE's advantage in balancing computational efficiency and forecasting performance, making it highly suitable for deployment in resource-constrained or real-time environments.

**Generalization of Parameter Sparsity.** To evaluate the generalizability of our sparsity-inducing strategy, we apply the L1 regularization to a diverse set of base models, including NLinear, iTransformer, and TimeMixer. As reported in Table 4, introducing L1 norm consistently improves the forecasting performance across all models, demonstrating that parameter sparsity is a universally beneficial inductive bias for long-term time series forecasting (LTSF). Specifically, we observe average MSE improvements of 4.4% on the NLinear model, 5.7% on iTransformer, and 2.3% on TimeMixer across the two datasets. These results suggest that L1-induced parameter sparsity not only benefits simple linear models but also effectively complements complex attention-based and mixer-based architectures. This further supports the claim that sparsity is a broadly applicable principle for enhancing generalization in LTSF tasks.

**Hyperparameter Analysis.** We study the effect of L1 regularization coefficient $\lambda$ on forecasting performance by varying its value and evaluating model accuracy using MSE and MAE, as shown in Table 3. A moderate value of $\lambda$ typically yields the best performance, indicating that inducing proper sparsity can enhance generalization, while overly strong regularization may suppress useful patterns.

To further investigate this behavior, we conduct additional experiments on more datasets, with detailed results reported in the supplementary material. Interestingly, we observe that for low-resource datasets such as ETTh1, ETTh2, and Exchange, relatively larger values of $\lambda$ lead to better performance, as they compensate for the lack of data-induced parameter sparsity. In contrast, on larger-

| Variants | ETTh1 (avg.) | | Exchange (avg.) | |
|---|---|---|---|---|
| | MSE | MAE | MSE | MAE |
| NLinear | 0.420 | 0.428 | 0.355 | 0.400 |
| + L1 | **0.402** | **0.417** | **0.345** | **0.391** |
| iTransformer | 0.439 | 0.448 | 0.360 | 0.404 |
| + L1 | **0.414** | **0.422** | **0.353** | **0.398** |
| TimeMixer | 0.427 | 0.441 | 0.381 | 0.416 |
| + L1 | **0.413** | **0.426** | **0.376** | **0.412** |
| Boost (%) | 4.4% | 3.9% | 2.0% | 1.6% |

Table 4: Generalization of parameter sparsity. Results are averaged from four forecasting horizons. "Boost" denotes the relative improvement after applying L1 regularization.

| Method | MACs | Params | Infer. Time | MSE |
|---|---|---|---|---|
| Informer | 3.97 G | 12.53 M | 70.1ms | 0.502 |
| Autoformer | 4.41 G | 12.22 M | 107.7ms | 0.254 |
| FEDformer | 4.41 G | 17.98 M | 238.7ms | 0.259 |
| FiLM | 4.41 G | 12.22 M | 78.3ms | 0.249 |
| PatchTST | 11.21 G | 6.31 M | 290.3ms | 0.214 |
| Pathformer | 8.69G | 7.92M | 156.94ms | 0.211 |
| PDF | 7.76G | 6.14M | 58.78ms | 0.199 |
| **CS-MoE** | **560M** | **1.69M** | **3.27ms** | **0.192** |

Table 5: Number of training parameters, MACs, inference time, and MSE of TSF models under look-back window = 512 and forecasting horizon = 720 on the large Electricity dataset.

scale datasets like Electricity and Traffic, smaller $\lambda$ values perform better since the model already exhibits natural sparsity driven by abundant data. This validates our hypothesis that L1 regularization serves as an effective sparsity prior under data scarcity, but can be relaxed when sufficient training data is available.

**Few Shot Learning.** To assess the effectiveness of sparsity in data-scarce scenarios, we conduct few-shot forecasting experiments by limiting the available training data to 10%, 20%, and 50% of the original training set. The validation and test sets remain unchanged, and all models are trained from scratch under each data condition using the same hyperparameters as in the full-data regime.

Table 6 presents the forecasting performance averaged over multiple horizons on the ETTh1 dataset. CS-MoE consistently achieves the best results across all few-shot settings. Notably, adding $\ell_1$ regularization to iTransformer and NLinear leads to moderate improvements, especially under 10% and 20% data, confirming that sparsity is a beneficial inductive bias under limited supervision. CS-MoE further improves upon this by integrating sparsity with modular routing and structured convolution.

| Data | 10% | 20% | 50% |
|---|---|---|---|
| iTransformer | 0.482 | 0.476 | 0.456 |
| iTransformer + L1 | 0.465 | 0.453 | 0.438 |
| NLinear | 0.451 | 0.438 | 0.423 |
| NLinear + L1 | 0.443 | 0.426 | 0.417 |
| **CS-MoE** | **0.429** | **0.417** | **0.403** |

Table 6: Few-shot forecasting performance (MSE) averaged over horizons on ETTh1.

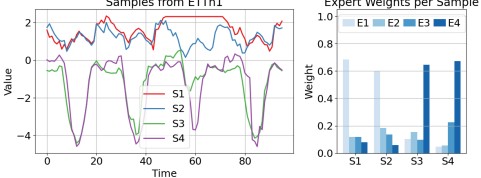

Table 7: Expert weights visualization on four ETTh1 samples. Left: standardized input series. Right: corresponding expert weights.

**Expert Weights Visualization.** Figure 7 shows the expert weights for four diverse samples from ETTh1. Samples with similar patterns (e.g., S1 and S2) activate similar experts (e.g., E1), while different patterns (e.g., S3 and S4) lead to distinct expert usage (e.g., E4), demonstrating that our model adaptively routes inputs to specialized experts based on temporal characteristics.

## 6 CONCLUSIONS

In this paper, we introduce CrossSparse-MoE, an efficient forecasting framework that mitigates overfitting in low-data regimes by novelly combining structural flexibility with adaptive sparsity. Motivated by a sparsity-oriented scaling law, the model leverages cross-channel convolutions for inter-variable modeling and employs a Mixture-of-Experts architecture with L1 regularization to learn compact and specialized representations. CrossSparse-MoE allows sparsity to emerge naturally through data-driven training. Experiments on real-world time series benchmarks show that our method achieves superior performance and extremely lightweight computational consumption.

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

## A  DATASET DESCRIPTIONS

Table 8: Dataset detailed descriptions. *Dim* denotes the number of variables per dataset, i.e., channels. *Frequency* represents the sampling interval of time points.

| Dataset | Dim | Timesteps | Frequency | Domain |
|---------|-----|-----------|-----------|--------|
| ETTh1&h2 | 7 | 17420 | Hourly | Electricity |
| ETTm1&m2 | 7 | 69,680 | 15-min | Electricity |
| Exchange | 8 | 7588 | Daily | Economy |
| Electricity | 321 | 26304 | Hourly | Electricity |
| Traffic | 862 | 17544 | Hourly | Transportation |
| Weather | 21 | 52696 | 10-min | Weather |
| ILI | 7 | 966 | Weakly | Disease |
| Solar | 137 | 52560 | 10-min | Energy |

- **ETT** consists of datasets with varying granularities, including two hourly-level datasets (ETTh1, ETTh2) and two 15-minute-level datasets (ETTm1, ETTm2). These datasets feature six power load variables and the target variable "oil temperature," spanning from July 2016 to July 2018.

- **Traffic** tracks hourly road occupancy on San Francisco freeways over the period of 2015–2016.

- **Electricity** provides hourly power usage data from 321 customers, collected from 2012 to 2014.

- **Exchange-Rate** contains daily foreign exchange rate data for eight countries, with records from 1990 to 2016.
- **Weather** consists of 21 weather parameters, including air temperature and humidity, recorded at 10-minute intervals throughout 2020 in Germany.
- **ILI** is sourced from the U.S. Centers for Disease Control and Prevention (CDC), documenting weekly instances of influenza-like illness from 2002 to 2021, including patient counts and ratios.
- **Solar** provides solar energy production data from 137 photovoltaic (PV) plants in Alabama.

**Drop Last Issue.**  Several studies Xu et al. (2024); Qiu et al. (2024) have highlighted the complications of using the "drop-last" setting during model evaluation. Specifically, enabling " drop_last=True" can lead to errors because of changes in the batch size of the test set. To mitigate these issues, we intentionally set the "drop_last=False" option for our experiments.

## B    HYPERPARAMETER SENSITIVITY

| $\lambda$ | Weather (avg.) | | Electricity (avg.) | | Traffic (avg.) | | ETTh2 (avg.) | |
|---|---|---|---|---|---|---|---|---|
| | MSE | MAE | MSE | MAE | MSE | MAE | MSE | MAE |
| 0.0 | 0.231 | 0.271 | 0.163 | 0.260 | 0.423 | 0.275 | 0.351 | 0.390 |
| 0.001 | 0.246 | 0.285 | 0.165 | 0.259 | 0.441 | 0.281 | 0.360 | 0.388 |
| 0.0001 | 0.235 | 0.270 | 0.160 | 0.263 | 0.421 | 0.276 | 0.351 | 0.384 |
| 0.00001 | **0.221** | **0.252** | 0.158 | 0.252 | 0.401 | 0.261 | **0.335** | **0.383** |
| 0.000001 | 0.225 | 0.261 | **0.154** | **0.247** | **0.394** | **0.258** | 0.342 | 0.394 |

Table 9: Effect of different L1 regularization coefficients $\lambda$ on forecasting performance. Results are averaged over all horizons on the Weather and Exchange datasets.

We conduct additional experiments on more datasets, with detailed results reported in the supplementary material. Interestingly, we observe that for low-resource datasets such as ETTh1, ETTh2, and Exchange, relatively larger values of $\lambda$ lead to better performance, as they compensate for the lack of data-induced parameter sparsity. In contrast, on larger-scale datasets like Electricity and Traffic, smaller $\lambda$ values perform better since the model already exhibits natural sparsity driven by abundant data. This validates our hypothesis that L1 regularization serves as an effective sparsity prior under data scarcity, but can be relaxed when sufficient training data is available.

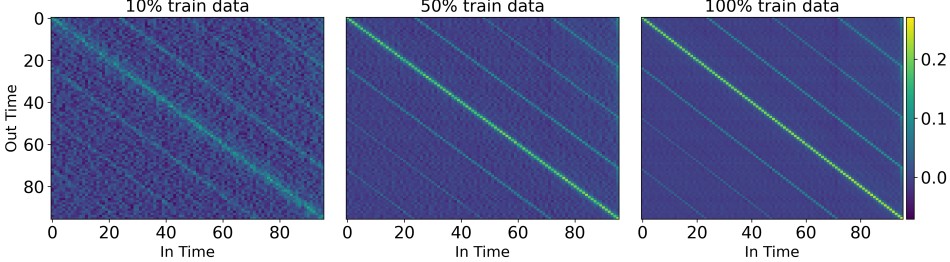

Figure 6: Visualization of weight sparsification with increasing data volume on the electricity dataset.

## C    EXPERIMENTAL RESULTS

We further extend our analysis to additional benchmark datasets by visualizing the weights of the model's linear projection layer (Figure 6, 8, 9, 7) and tracking two sparsity metrics (Figure 10, 12,

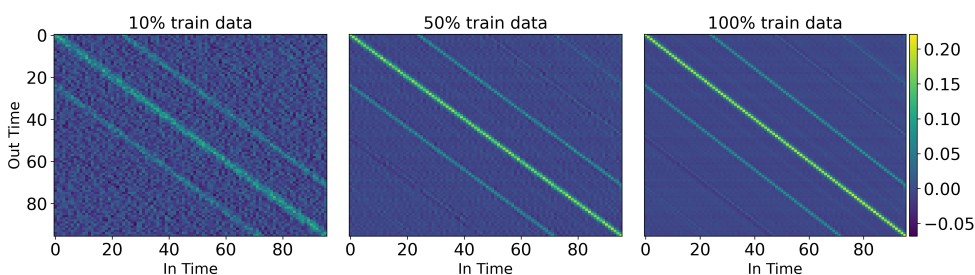

Figure 7: Visualization of weight sparsification with increasing data volume on the traffic dataset.

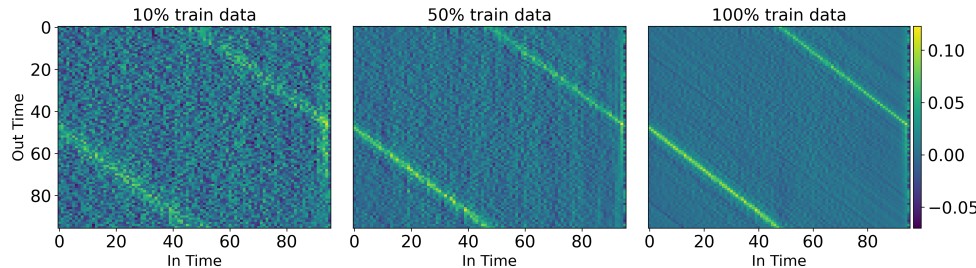

Figure 8: Visualization of weight sparsification with increasing data volume on the ETTm1 dataset.

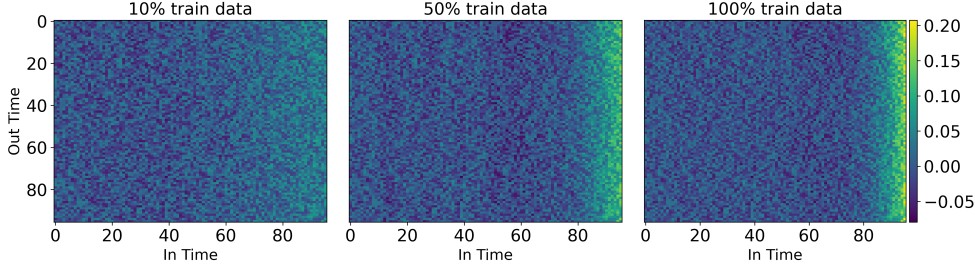

Figure 9: Visualization of weight sparsification with increasing data volume on the exchange rate dataset.

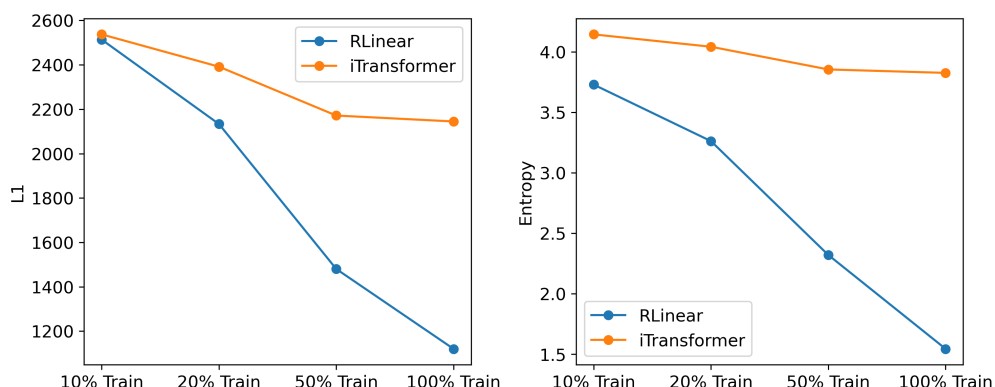

Figure 10: Metrics for parameter sparsification on electricity dataset.

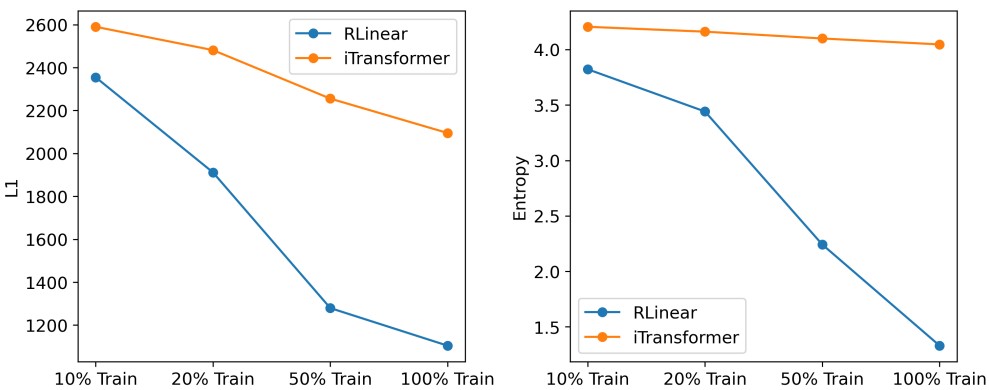

Figure 11: Metrics for parameter sparsification on traffic dataset.

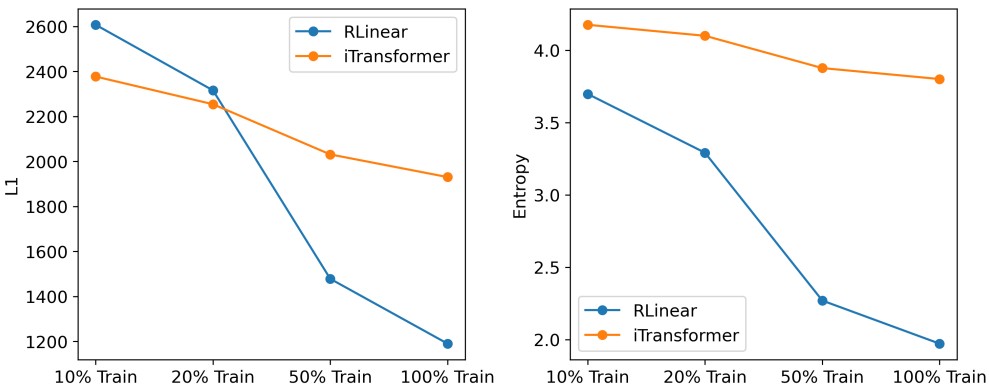

Figure 12: Metrics for parameter sparsification on ETTm1 dataset.

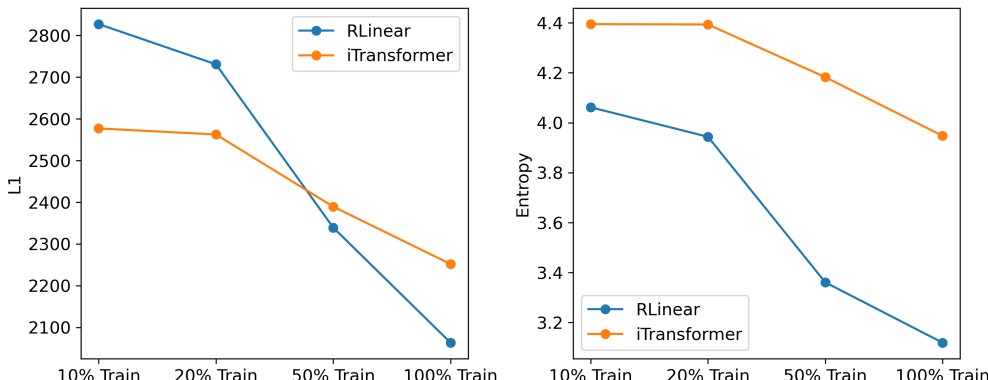

Figure 13: Metrics for parameter sparsification on ETTh1 dataset.

13, 11) as the amount of training data increases. A consistent trend emerges: as the data volume grows, the model exhibits increased parameter sparsity, with weights corresponding to less important features gradually approaching zero. Additionally, weight patterns differ significantly across datasets. For example, the Exchange dataset displays weak periodicity, with weights primarily focused on the most recent timesteps, whereas the ETTm1 dataset shows clear periodicity with a cycle length of 96.

## C.1 USE OF LLMs

During the preparation of this manuscript, we used the OpenAI ChatGPT (GPT-5) large language model as an assistant for language refinement, grammar correction, and style improvement. The model was also employed for suggesting alternative phrasings and generating draft outlines of certain sections, which were subsequently reviewed, verified, and substantially revised by the authors. All technical content, experiments, analyses, and conclusions presented in this paper were conceived, implemented, and validated solely by the authors. The authors take full responsibility for the accuracy and integrity of the manuscript's content.

