# OpenReview forum: "CrossSparse-MoE: Adaptive Sparsity and Cross-Channel Expert Routing for Time Series Forecasting"
_ICLR.cc/2026/Conference — Submitted to ICLR 2026_

### Official Review · Reviewer_ZpDZ · 2025-10-28

**Soundness:** 3
**Presentation:** 3
**Contribution:** 2
**Rating:** 4
**Confidence:** 4

**Summary:**

This paper introduces CrossSparse-MoE, a lightweight time series forecasting framework that combines cross-channel convolutional embeddings with a Mixture-of-Experts architecture. It identifies a sparsity-oriented scaling law showing that model parameters become naturally sparser as training data increases. The method leverages L1 regularization to promote adaptive sparsity, improving generalization in low-data regimes. Extensive experiments across ten benchmarks demonstrate state-of-the-art performance and high computational efficiency.

**Strengths:**

The paper presents a observation of a sparsity-oriented scaling law in time series forecasting, offering a fresh perspective on data-driven regularization. The proposed CrossSparse-MoE framework effectively combines adaptive sparsity with modular expert routing, achieving strong results with high efficiency. The work is clearly written, and demonstrates consistent improvements across diverse benchmarks, highlighting both its technical quality and practical relevance.

**Weaknesses:**

The main limitation lies in the theoretical grounding of the proposed sparsity-oriented scaling law, which is described heuristically without formal derivation or quantitative fitting. The architectural novelty is moderate, as CrossSparse-MoE builds upon established concepts from MoLE and Time-MoE with added L1 regularization. The ablation analysis could be expanded to study the number of experts, gating behavior, and α interpolation dynamics. Finally, robustness experiments under distribution shifts or noise perturbations would strengthen the claim that adaptive sparsity improves generalization.

**Questions:**

1. Can the authors provide quantitative evidence for the proposed sparsity-oriented scaling law, such as fitting the parameters ($C_1$, $C_2$, $\beta$ ) in Eq. (3) or showing trends across datasets?

2. How sensitive is the model to the number of experts and the gating network capacity? Would larger expert pools or deeper gates improve performance or cause overfitting?

3. Could the authors analyze expert specialization, for instance by clustering input patterns or reporting activation diversity, to better support claims of modularity and interpretability?

4. How does CrossSparse-MoE perform under domain shift or noisy sensor conditions, where sparsity might enhance robustness?

---

### Official Review · Reviewer_izTy · 2025-10-31

**Soundness:** 2
**Presentation:** 2
**Contribution:** 2
**Rating:** 2
**Confidence:** 4

**Summary:**

The paper proposes CrossSparse-MoE, a time series forecasting model that combines adaptive sparsity and MoE routing. It is based on a sparsity-oriented scaling law, showing that model sparsity naturally increases with more data. The model uses cross-channel embeddings and temporal expert routing with L1 regularization to improve generalization under limited data. Experiments show state-of-the-art accuracy and efficiency across several benchmarks.

**Strengths:**

1. The motivation, from the perspective of parameter sparsity, is quite interesting.

**Weaknesses:**

1. The Introduction is not written rigorously. In line 38, it is stated that most existing works rely on high-quality labeled datasets, which is not accurate. In time series forecasting, training data usually comes from real-world time series, where part of the sequence is used as input and another part as the label. Therefore, there is no issue of label quality. In addition, in the second paragraph (lines 41–42), the cited papers mainly aim to improve model efficiency, not robustness under limited data.

2. The preliminary study on parameter sparsity is not convincing enough. The analysis mainly focuses on linear models, lacking experiments or discussions on more complex architectures such as Transformer-based, CNN-based, or MLP-based models.

3. The paper lacks novelty, as the MoE framework has already been widely used in many existing works.

4. The writing is not concise, and the readability is relatively poor. For example, the content in Section 4.1 reflects well-known domain knowledge and could be moved to the appendix. Furthermore, the connection between the preliminary study and the proposed method is not clearly established.

**Questions:**

1. Could you explain why high-quality labeled datasets are often limited?

2. Which figure does Figure 3.3 refer to?

3. In Figure 3, what does the vertical axis represent? Why does RLinear drop significantly while iTransformer shows almost no change?

4. How are the conclusions from the preliminary study integrated into the MoE framework?

---

### Official Review · Reviewer_K7nC · 2025-11-01

**Soundness:** 3
**Presentation:** 4
**Contribution:** 3
**Rating:** 6
**Confidence:** 3

**Summary:**

This paper introduces CrossSparse-MoE, a lightweight time series forecasting framework designed to address overfitting in low-data regimes through dynamic parameter sparsity. The main csontribution in this paper is the identification of a empirical observation called "sparsity-oriented scaling law" where the model parameters naturally become sparser as training data increases, even without explicit regularization. Building on this insight, the authors propose a hybrid architecture combining: (1) cross-channel convolutional embeddings for inter-variable dependencies, and (2) a Mixture-of-Experts (MoE) module with L1 regularization to encourage adaptive sparsity. Experiments on several benchmark datasets demonstrate state-of-the-art performance, particularly in low-resource settings.

**Strengths:**

The main strengths of the paper include,
1. The sparsity-oriented scaling law observation is interesting and well-documented across multiple datasets (ETTh2, Weather, Electricity, Traffic, etc.) and the authors provided clear evidence of increasing weight sparsity with training data volume in their plots.
2. The proposed CS-MoE achieves best performance in several benchmark datasets compared against strong baselines (PatchTST, DLinear etc) while achieving super efficiency of just 3.27ms inference time vs competitors and having only 1.69M params (orders of magnitude smaller than baselines).
3. The paper includes clear ablation studies for 1. Contribution of individual components of the CS-MoE framework. 2. Sensitivity of \lambda parameter in L1 regularization.

**Weaknesses:**

The main weakness of the paper include,
1. The sparsity-oriented scaling law observation is only validated on linear projection layers of simple models (RLinear, iTransformer's linear components). No analysis is conducted for other layer types such as attention layers, non-linear MLP layers, or convolutional layers. This severely limits the applicability of the claimed "scaling law" to broader model architectures. The paper claims this is a general phenomenon but only demonstrates it for a specific layer type.
2. The paper claims about state-of-the-art performance in few-shot settings of the proposed CS-MoE framework but it does not compare against recent pre-trained time series models (TimesFM, Chronos, Lag-Llama, Moment, etc.) which are specifically designed for low-data scenarios. So, this claim cannot be justified unless further evidence is provided.
3. The paper claims to "prevent model overfitting for sparse time series" but provides no specific anecdotes (or) metrics for sparse/irregular time series.

**Questions:**

1. What constitutes a "sparse time series" in your context? Can you provide metrics/anecdotes on datasets with missing values, irregular sampling, or high # of zero-values?
2. What is the sensitivity to the number of experts?
3. Can you validate the sparsity-oriented scaling law for other layer types apart from linear layers?
4. Can you compare the benchmark with some of the pre-trained time series models as they are specifically designed for zero-shot scenarios?

---

### Official Review · Reviewer_wy7n · 2025-11-01

**Soundness:** 1
**Presentation:** 2
**Contribution:** 2
**Rating:** 2
**Confidence:** 4

**Summary:**

This paper attempts to explain why model performance improves after training on large amounts of data from the perspective of weight sparsity, which is quite insightful. Based on this insight, it designs a lightweight sparse model using L1 regularization, which shows good prediction accuracy and low model overhead on multiple datasets.

**Strengths:**

This paper attempts to explain why model performance improves after training on large amounts of data from the perspective of weight sparsity, which is quite insightful. Based on this insight, it designs a lightweight sparse model using L1 regularization, which shows good prediction accuracy and low model overhead on multiple datasets

**Weaknesses:**

1. It seems natural that after the model is trained on a larger amount of data, it learns to extract important patterns and reduce the weights of redundant features, making the learned patterns clearer. Similar patterns have also been mentioned in other papers. The connection to sparsity here seems a bit forced.

2. The innovation is limited. The method in the paper is simply a combination of L1 regularization and multiple dense multi-predictor experts applied to the prediction task. The use of a mixture of experts is very similar to that used in MoLE.

3. The experiments in the paper are not very convincing. For example, the ablation study was only conducted on the relatively small ETTh1 and Exchange datasets. Additionally, some designs, such as alpha, were not ablated.

4. Figure 5 is not very clear. The MOE predictor does not correspond to the steps in Figure 5.

**Questions:**

Please refer to Weaknesses.

---

### Meta-Review · Area_Chair_iGef · 2026-01-07

**Summary:**

This submission proposes CrossSparse-MoE, motivated by an empirical “sparsity-oriented scaling” observation (that weights become sparser as training data grows), and implements a lightweight forecasting model that combines cross-channel convolutions with MoE routing under L1 regularization. Reviewers found the observation interesting and appreciated the strong efficiency claims and competitive results on several benchmarks (one reviewer rated it above the acceptance threshold). At the same time, the dominant concerns were about (i) overstated generality of the “scaling law,” since evidence is largely limited to linear layers / simple linear components, with no convincing validation on attention/MLP/conv layers; (ii) limited novelty, as the method is seen as a fairly direct combination of L1 sparsity + MoE, similar in spirit to prior MoE forecasting designs; and (iii) incomplete experimental support, including limited ablations (for example, number of experts, α/interpolation, gating behavior) and missing comparisons against pretrained/zero-shot time-series models designed for low-data regimes (TimesFM, Chronos, Lag-Llama, MOMENT). Reviewers also flagged clarity issues in the motivation and figures, and asked for robustness under shifts/noise and a clearer definition of “sparse time series.”

**Reviewer Scores:**

Given the current record, the two reject reviewers would likely remain rejecting unless the authors substantially strengthen the evidence that the sparsity phenomenon extends beyond linear projections and clarify the novelty relative to prior MoE forecasting work. The borderline reviewer might increase modestly if the authors provided quantitative fits/validation for the scaling claim, expanded ablations, and added robustness tests. The supportive reviewer could stay similar, but might also drop slightly if the missing pretrained baselines and the limited scope of the scaling-law evidence remain unaddressed.

---

### Decision · Program_Chairs · 2026-01-26

Reject